# Training for Pediatric Sepsis—A Medical Education Perspective and Potential Role of Artificial Intelligence

**DOI:** 10.3390/children12111542

**Published:** 2025-11-14

**Authors:** Spyridon Karageorgos, Owen Hibberd, Dennis Ren, Yasmin Hornsby, Damian Roland, Ioannis Koutroulis

**Affiliations:** 1Faculty of Medicine and Dentistry, Blizard Institute, Queen Mary University of London, London E1 2AT, UK; s.karageorgos@qmul.ac.uk (S.K.); o.hibberd@nhs.net (O.H.); 2Pediatric Emergency Department, Aghia Sophia Children’s Hospital, 11527 Athens, Greece; 3Emergency and Urgent Care Research in Cambridge (EURECA), PACE Section, Department of Medicine, Cambridge University, Cambridge CB2 0QQ, UK; yh471@cam.ac.uk; 4Division of Emergency Medicine, Children’s National Hospital, Washington, DC 20012, USA; dmren2@childrensnational.org; 5School of Medicine and Health Sciences, The George Washington University, Washington, DC 20052, USA; 6SAPPHIRE Group, Population Health Sciences, Leicester University, Leicester LE1 7RH, UK; dr98@leicester.ac.uk; 7Pediatric Emergency Medicine Leicester Academic (PEMLA) Group, Children’s Emergency Department, Leicester Royal Infirmary, Leicester LE1 5WW, UK

**Keywords:** pediatric sepsis, medical education, artificial intelligence

## Abstract

Pediatric sepsis is a major cause of morbidity and mortality worldwide, with outcomes dependent on timely recognition and rigorous management. As clinical management of pediatric sepsis depends on early recognition and initial therapeutic steps, targeted educational materials for healthcare workers in these early phases of care are warranted. Findings of this review highlight and compare the role of traditional educational methods (e.g., lectures) to alternative teaching methods (e.g., use of virtual reality) in educating healthcare workers about pediatric sepsis. Overall, there is a gradual shift from traditional, teacher-centered, transmissive teaching methods to more collaborative, reflective, and learner-centered approaches. These pedagogical approaches, despite some potential limitations, offer opportunities to use technological enhancements and Artificial Intelligence (AI) to enhance teaching and learning across various methods.

## 1. Introduction

Pediatric sepsis is a major cause of morbidity and mortality worldwide, with outcomes dependent on timely recognition and rigorous management [1]. While the majority of children with sepsis initially present to emergency care settings, a large proportion of Emergency Department (ED) visits are due to febrile children with minor viral illnesses. Distinguishing these benign cases from true sepsis continues to pose a significant clinical challenge [2].

Training in pediatric sepsis is crucial for the prompt identification of cases, timely delivery of treatment through adherence to guidelines, and the standardization of care, definitions, and research across centers. Many studies highlighted a lack of consensus on various definitions and identified that training could improve standardization, resulting in improved care and more valid study results. For example, a quality improvement project identified difficulties with standardizing “time zero” for sepsis onset, recommending international efforts to unify definitions [3].

Moreover, a quality improvement project that included 19 hospitals, focused on ED care, provided training on the application of sepsis criteria and standardized screening tools, and found that improved time to first clinical assessment, fluid bolus and antibiotics was associated with a decrease in 30-day all-cause in-hospital mortality [4]. This multisite study also demonstrated that training can be successfully delivered across sites and results in improvements in multiple hospitals. However, not all areas saw improvement. For example, there was no significant increase in the administration of antibiotics within 1 h, which demonstrates that “one size does not fit all” and that more specific training focused on the various aspects of sepsis care needs to be developed. A 2018 Delphi process study [5] established pediatric sepsis educational goals as presented in Table 1.

Educational training programs (e.g., simulation, virtual reality simulation) focused on training pediatric and pediatric emergency trainees are necessary to improve adherence to sepsis clinical pathways. The use of Artificial Intelligence (AI) to enhance the learning experience has become an emerging theme in medical education. However, there is scarce data regarding the use of AI in pediatric sepsis education.

This review focuses on assessing current knowledge regarding pediatric sepsis education and discussing the potential use of AI to enhance pediatric sepsis education.

## 2. Methods

For this narrative review, a literature search on Medline up to March 2025 was performed with the aim of identifying different teaching methods on pediatric sepsis published in English. Search terms included “pediatric sepsis”, “medical education” and “artificial intelligence (AI)”. Articles were chosen based on their focus on the use of different medical education training programs and the use of artificial intelligence (AI) in educational efforts centered around pediatric sepsis. Authors subsequently performed a thematic analysis of the findings [6]. Thematic analysis included the following categories:Lectures;Seminars;E-learning;Hybrid learning;Technology-enhanced learning (gamification, virtual reality (VR) and augmented reality (AR) simulation;Artificial intelligence;Barriers to implementation of artificial intelligence in sepsis education.Pediatric sepsis definitions vary across different studies:International Pediatric Sepsis Consensus Conference (IPSCC) 2005 defined sepsis as Systematic Inflammatory Response Syndrome (SIRS) in the presence of suspected or proven infection [1,2].Sepsis-3 criteria defined sepsis as Sequential Organ Failure Assessment (SOFA) Score ≥ 2 and presence of infection [1,2].Phoenix criteria for pediatric sepsis in 2024 defined sepsis as Phoenix Sepsis Score ≥ 2 points and suspected infection [1,2].

## 3. Results

### 3.1. Lectures and Seminars

Traditionally, sepsis education was conducted in classroom settings through lectures and seminars (Figure 1).

### 3.2. Lectures

Teaching through lectures on sepsis education has relied on transmissive, teacher-centered learning models [7,8,9]. This generally involves lectures and classroom-based, didactic teaching through the delivery of presentations [7,8]. The ethos underlying this approach is that knowledge is imparted to the learner by the educator through a presentation [7,8]. These are delivered live and in person, providing the advantage of opportunities to interact with the educator and a community of peers, while also being time-efficient and allowing a standardized knowledge set to be delivered to a large cohort of learners [7,8]. Additionally, this format allows students to learn professionalism by observing their peers and the educator [10]. This also represents the most used teaching modality in low- and middle-income country (LMIC) settings [11,12]. These are all parts of a pedagogic approach in which teachers decide what, how and when something will be learned [13]. However, disadvantages include the educator setting the pace, which often results in limited dynamic interaction between the learner and educator, inducing a passive learning approach and a one-way transfer of knowledge [7,14]. In the context of sepsis education, this can enable educators to deliver a significant volume of the standardized content and essential knowledge recommended by consensus guidelines [5]. For example, a survey involving 549 medical students from two medical schools in Queensland assessed their knowledge of sepsis [15]. Most respondents (349/549, 63.6%) reported receiving formal education on sepsis predominantly through didactic lectures [15]. Less than half of the students were aware of the significance of timely interventions and the national burden of sepsis [5,15]. The survey found that many medical students felt unprepared to recognize and manage septic patients, indicating a need for a standardized curriculum focused on sepsis education and more active teaching methods [15].

### 3.3. Seminars

Seminars are a face-to-face, small-group method of teaching that promotes more active learning [7,8]. During seminars, learners work in small groups to engage in learner-oriented discussions focused on specific questions or cases, guided by the educator [8]. Seminars have the advantage of allowing multidirectional interaction among the learners and the educator, enabling the learner to better develop skills such as collaborative teamwork, scientific questioning, and debate [7,8]. For learning about sepsis, this is particularly advantageous as it emphasizes teamwork and clinical reasoning, which can help students to understand and apply knowledge related to sepsis [5,7]. In contrast to lectures, a potential disadvantage of this method relates to its operationalization and logistics, as seminars require greater resources, learner buy-in, and place a more intensive burden on the learner due to the active nature of participation [7,8]. A meta-analysis that included 16 randomized controlled trials with a total sample of 1122 medical students explored the effects of seminar teaching methods compared to lecture-based learning in medical education [8]. This meta-analysis found that the seminar teaching method significantly improved knowledge scores in practice courses but had no substantial effects on theory courses [8]. Given the previous shortcomings in sepsis education, which have left learners underprepared to recognize and manage it, a practical approach arguably allows this knowledge to be better translated into the clinical environment and supports the transition from medical student to doctor [5,8,15]. However, the meta-analysis also indicated that lecture-based learning is effective in teaching theoretical elements, and for sepsis education, this might include teaching on organisms and antibiotic regimes [5,8].

### 3.4. E-Learning and Hybrid Learning

Over the past decade, there has been a growing interest in developing e-learning and hybrid learning methods, which have increasingly been utilized for the delivery of sepsis education (Figure 2).

### 3.5. E-Learning

The emergence of e-learning reflects technological advancement and the transition of medical education from a teacher-centered, transmissive method of learning to a more collaborative and reflective, learner-centered model [7,14,16,17]. E-learning uses online learning, technology, and other electronic media to facilitate knowledge transfer [17]. This approach is supported by the principles of connectivism learning theory, which views knowledge as a dynamic entity spread through technology-enabled networks that foster interactions between individuals and society generally [16,18]. Although e-learning has already played a leading role in post-graduate learning and continuing professional education, the COVID-19 pandemic has created a need for e-learning to be rapidly integrated into undergraduate medical curricula [7,16]. COVID-19 disrupted education worldwide, and to prevent the spread of infection, face-to-face learning was halted, leading many organizations to transition to online learning to ensure that the education of medical students continued [7,16,19]. Evidence exploring e-learning and the perspectives of both learners and educators has demonstrated varied results [7,16,19]. E-learning offers advantages such as flexibility, accessibility, and the capacity to deliver a standardized level of content [5,7]. It also incorporates the principles of andragogy, reflection, and self-evaluation, guiding learners through the learning cycle with prompts and tasks [20,21,22]. Andragogy focuses on critical thinking and the enhancement of knowledge in adult learners via application of knowledge in real-life scenarios [13]. In the context of sepsis education, e-learning can facilitate the delivery of core content and knowledge while allowing learners to engage with clinical vignettes and hypothetical scenarios, providing them with opportunities for self-evaluation and reflection [5,16,17]. However, this format does not include face-to-face interaction, which may better replicate the complexities of the clinical environment, and the efficacy of e-learning is highly variable depending on the format, application, and available resources [7,16,17]. Satisfaction with e-learning has been reported to be much lower in the LMIC setting [10,12,19,23,24,25]. For example, a survey of 824 medical students in Pakistan observed that 668/824 (81%) expressed overall dissatisfaction with e-learning, while 631/824 (77%) faced technical issues such as internet accessibility and a lack of IT-related skills [12]. It is recommended that for e-learning to be applied effectively, resources should be sufficient, the learning should promote critical thinking and reflection, and it should incorporate principles to optimize the format of content [17]. For instance, Richard Mayer’s Theory of Multimedia Learning and Universal Design for Learning principles describe how to optimize teaching and learning by enhancing learner perception and comprehension through the way learning is presented [14,26,27].

### 3.6. Hybrid Learning

Utilizing a combination of face-to-face and online learning modalities, blended learning enables students to employ e-learning technology to engage with various self-paced activities outside the classroom before attending seminars or lectures [28]. This ‘flipped classroom’ model allows class time to concentrate on clinical reasoning, application, and more nuanced or challenging topics, which may be particularly beneficial for sepsis education [14,29]. Face-to-face sessions can also be enhanced by technology, such as interactive digital tools and quizzes, to improve student engagement [17,28]. Examples of free, open-access medical education resources that can be used to support blended learning for sepsis education include the Pediatric Education and Advocacy Kit (PEAK): Sepsis, from the Emergency Medical Services for Children (EMSC), and the SIRS, Sepsis, and Shock module from Don’t Forget The Bubbles [30,31]. The efficacy of blended learning has been examined in the literature [7,16,28]. For instance, a meta-analysis assessing the effectiveness of blended learning compared to traditional learning found that blended learning consistently outperformed traditional methods on knowledge outcomes in medical education (standard mean difference 1.07, 95% CI 0.85 to 1.28) [28]. In India, a qualitative Strengths, Opportunities, Aspirations, Results (SOAR) analysis of a large group of medical and dental students from a single university investigated the strengths and barriers of blended learning in a low- and middle-income country (LMIC) setting [10]. Overall, blended learning was deemed to facilitate greater engagement during learning and support self-directed learning both directly and indirectly; however, barriers such as internet connectivity remained, and some students expressed a preference for conventional face-to-face teaching [10]. Ultimately, blended learning is an effective approach to maximizing the benefits of both face-to-face and online learning [7,16,28]. This method holds potential in an LMIC and may serve as a foundation for future utilization of AI in medical education.

### 3.7. Technology-Enhanced Learning and AI

With recent advances in technology, newer and more innovative educational methods have emerged as modes for delivering sepsis education. These include the gamification of teaching and simulation, which continue to evolve rapidly with the advent of AI (Figure 3).

### 3.8. Gamification

Gamification refers to the use of game-design elements in non-game settings [32]. These elements can include scoring systems, leaderboards, rewards, competition, and other social elements. There has been growing interest in the use of gamification in health professions education, which can include sepsis training [32].

One example is the online educational game Septris, developed at Stanford University. Septris uses time-sensitive, case-based scenarios where learners manage fictional patients presenting with varying severities of sepsis. Players receive feedback and score bonuses for evidence-based decisions. The game reinforces key clinical principles such as early recognition, fluid resuscitation, and antibiotic administration. Users of Septris showed significant improvement in sepsis-related knowledge and self-assessed competence after just 20 min of game play [33].

However, there are limitations to the systematic review literature around gamification. First, there is no consensus as to what exactly qualifies as a “game”, which may have resulted in studies being missed or improperly excluded. Most studies around the use of gamification were descriptive, involved small groups of learners in the United States and Canada. Third, studies did not report any potential negative outcomes associated with gamification, and it is unclear which mechanisms or elements of gamification are responsible for learning effects.

The application of AI in gamified platforms can allow for more dynamic and tailored game scenarios based on a learner’s performance, adjusting difficulty or targeting specific learning objectives of each game. This would include the development of clinical scenarios for pediatric sepsis, assessment of timely interventions and feedback on overall performance.

### 3.9. Simulation

Simulation-based learning is a cornerstone in a wide array of health professions education, used to teach technical and non-technical skills [34]. Simulation supports experiential learning, allowing learners to make and correct mistakes without jeopardizing patient safety. It also promotes retention and transfer of clinical skills to real-world settings [34].

Pediatric sepsis simulation allows learners to develop skills such as quick recognition of septic shock, rapid fluid resuscitation, and the administration of antibiotics. It also allows for the practice of team dynamics and clear communication. A study by Dugan et al. investigated the effect of repeated sepsis simulation on pediatric residents. Participants exposed to more simulations were more accurate in diagnosing septic shock and administering fluids and antibiotics compared to those with a single simulation [34]. The optimal frequency of conducting sepsis simulation for knowledge and skill retention is not currently known.

AI can potentially augment simulation by managing scenario flow in real-time based on participant actions. AI can assist in creating realistic clinical scenarios on pediatric sepsis and provide feedback regarding timely interventions. This would assist in assessing the time and adherence to sepsis guidelines in simulation scenarios. For example, AI coaching may be able to detect moments of hesitation or incorrect clinical choices and offer guidance. By capturing these key moments, AI can also help generate points for discussion during simulation debriefing or provide individualized feedback to each learner, capturing nuanced performance data and delivering intelligent debriefing insights [35]. In a recent study, a structured approach using mind mapping and in situ simulation training in ED nurses resulted in a statistically significant increase in identifying sepsis earlier and promoted the Hour-1 bundle treatment in sepsis patients [36].

### 3.10. Virtual and Augmented Reality Simulation

There is growing interest in virtual and augmented reality (VR/AR) technologies for medical education. These modalities provide immersive environments where learners can interact with digital patients or visualize pathophysiological processes [37]. PediSepsisAR is an augmented reality application that can be used in conjunction with traditional pediatric sepsis simulation [38]. It overlays a digital circulatory system onto a traditional simulation manikin, allowing learners to visualize changes in perfusion during fluid resuscitation. Although it did not significantly change fluid administration times compared to non-augmented simulation, users reported enhanced awareness of the patient’s hemodynamic status. Although novel, it will be important to continue to evaluate the cost-effectiveness of these tools in specific applications.

AI can enhance VR/AR experience for learners by using natural language processing abilities to allow virtual avatars to respond realistically to medical interventions and commands. Natural language processing ability of AI can allow learners to interact directly with the virtual avatars and have conversations. This was also shown in a mixed methods study that used an AI-enabled VR simulator (VRS) and showed enhanced learning and improved multidisciplinary communication between nursing students and an AI medical doctor [39]. This would be key in pediatric sepsis education, as communication is key in providing timely interventions.

### 3.11. Barriers to the Implementation of AI in Sepsis Education

While AI-enhanced tools in pediatric sepsis education may be enticing, it is important to recognize that implementation may be limited by several factors. For example, significant barriers include the cost and infrastructure requirements. Many of these platforms require substantial initial investment and ongoing maintenance. This disparity is particularly pronounced in low- and middle-income countries (LMICs), where access to reliable internet, computing power, and educational software is limited. This may widen the global educational gap, where advanced digital tools are more easily accessible in higher-income countries. Efforts toward global equity in pediatric sepsis education should prioritize scalable and cost-effective solutions [40]. On the other hand, high-income countries have access to high-fidelity simulation resources and infrastructure resources that offer realistic clinical scenarios, enhancing clinical decision making and improving communication between multidisciplinary teams [41]. In a recent systematic review regarding the effectiveness of low-cost, technology-enhanced simulation training, data showed that low-cost, technology-enhanced simulation tools may serve as an efficient strategy. This may reduce disparities in access to high-quality training between high-income countries and LMICs [42].

Additionally, there is currently a lack of structured training programs that teach clinicians how to effectively interpret and apply AI output in medical education and clinical practice. Moreover, prompt use of AI in medical education requires a multidisciplinary collaboration of multiple teams, including but not limited to clinicians, data scientists, and ethics committees [43]. Also, a big problem nicely described in a narrative review is that there is the potential that clinical supervisors may be less experienced than students in the use of AI. This underlines the importance of training the trainers on AI use in medical education [44].

Another major barrier is the ethical considerations behind the use of AI in medical education and sepsis education. Currently, there is limited training in the ethical and appropriate use of AI in medical education. In fact, a recent scoping review on teaching AI ethics in medical education highlighted the need for incorporating AI teaching into the medical curriculum and also the importance of training future generations of healthcare professionals on the prompt and ethical use of AI in medical education [45].

Lastly, as AI use in healthcare simulation grows, so does its environmental footprint. Training and operating AI models often require significant energy and data processing resources, contributing to carbon emissions. Developers and educators should consider the environmental impact of implementing these new technologies [46].

### 3.12. Limitations

As a narrative review, no quantitative analysis was performed. Also, although pediatric sepsis educational goals have been suggested, there were no studies evaluating how these goals changed the pediatric sepsis educational landscape. Future studies should focus on providing data on the impact these educational goals had on pediatric sepsis education.

## 4. Conclusions

Sepsis education is vital for improving the management of children with sepsis and reducing related mortality. Overall, this review presents a shift from traditional, teacher-centered, transmissive teaching methods to a more collaborative, reflective, and learner-centered approach, with benefits and disadvantages for both. These pedagogical approaches both offer opportunities to use technological enhancements and AI to support teaching and learning across various methods. Challenges exist in promoting fair and widespread AI adoption in education; therefore, future research should aim to identify the most effective and equitable methods to utilize it.

## Figures and Tables

**Figure 1 children-12-01542-f001:**
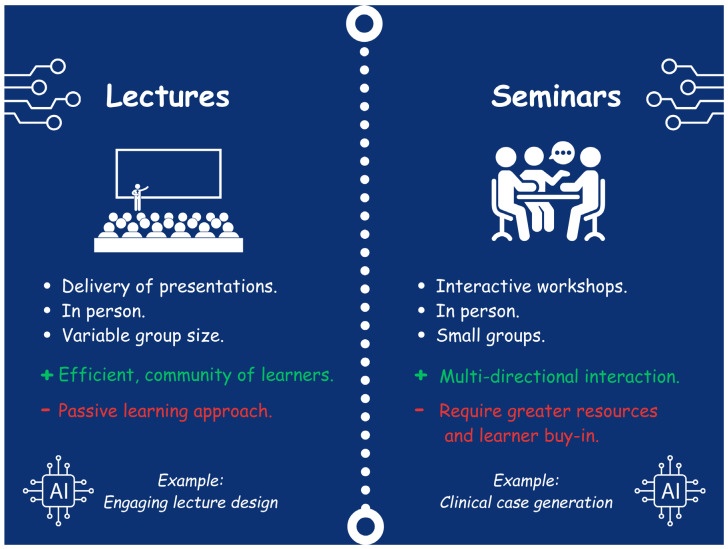
Lectures and Seminars.

**Figure 2 children-12-01542-f002:**
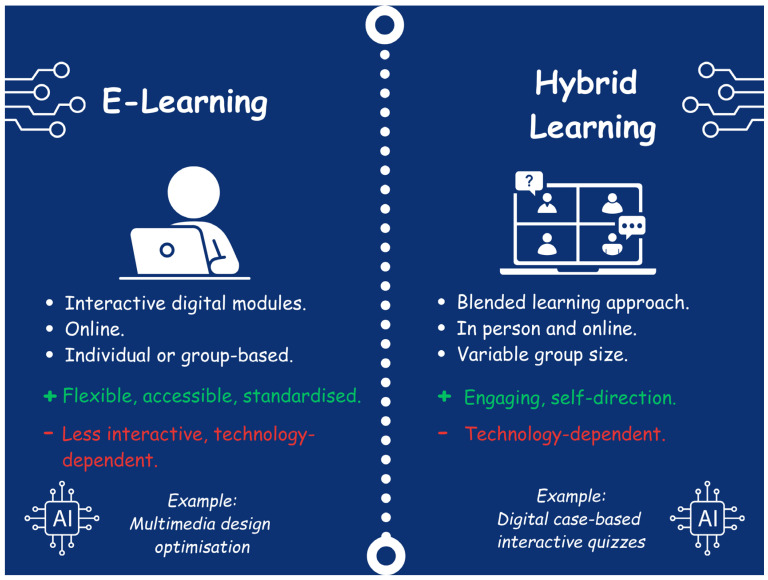
E-learning and Hybrid Learning.

**Figure 3 children-12-01542-f003:**
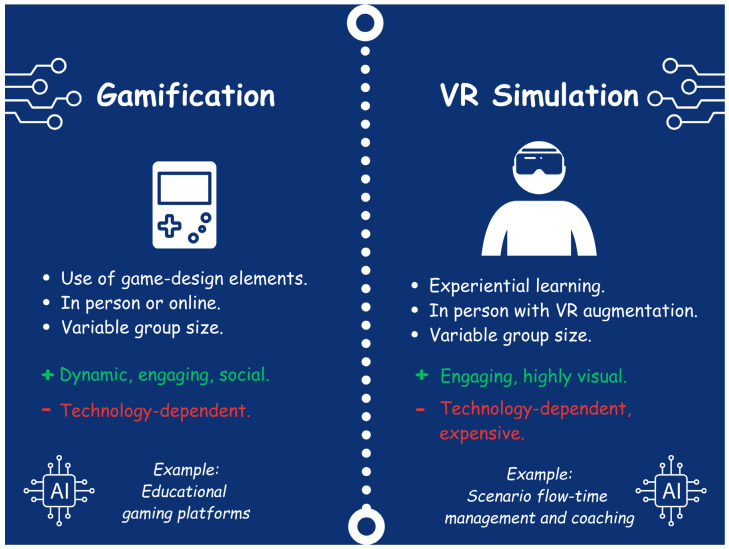
Gamification and Virtual Reality (VR) Simulation.

**Table 1 children-12-01542-t001:** Pediatric Sepsis Educational Goals.

1. Recognize possible septic shock signs: tachycardia, abnormal capillary refill, abnormal mental status, systolic hypotension, widened pulse pressure, elevated lactate
2. Initiate early IV access, with large diameter and short length preferred
3. Initiate IV fluid resuscitation with the appropriate agent and volume
4. Deliver IV fluids rapidly using rapid delivery system and specifically not with an IV pump
5. Deliver appropriate antibiotics within 180 min of sepsis recognition
6. Recognize appropriate responses to fluid therapy by normalization of or improvement in heart rate, blood pressure, capillary refill time, lactate, and/or urine output
7. Continue fluid resuscitation and prepare for potential vasoactive agent requirement when septic shock persists despite appropriate initial rapid fluid resuscitation
8. Recognize fluid overload during resuscitation and appropriately choose adjunctive therapy
9. Initiate vasoactive infusion through any access available when hypotension is present despite appropriate rapid fluid resuscitation
10. Obtain blood culture prior to giving antibiotics
11. Recognize and correct hypoglycemia
12. Appropriate pressor choice
13. Consider corticosteroids in catecholamine-resistant shock. Treat and test appropriately
14. Recognize and correct hypocalcemia

Data adapted from [5].

## Data Availability

Not applicable.

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
