# Peer review of "Training for Pediatric Sepsis—A Medical Education Perspective and Potential Role of Artificial Intelligence"

_children, 2025, doi:10.3390/children12111542_

Round 1

Reviewer 1 Report

Comments and Suggestions for Authors

Thank you for the opportunity to review this paper.

The paper aims to provide a comprehensive review of medical education specifically tailored to the teaching of pediatric sepsis and in the emergency room. However, the paper seems to read more for general medical training strategies than sepsis education specifically and I think the paper either needs to shift goals to define general education strategies and the role for AI or get more detailed with regards to sepsis education.  

AI has essentially been introduced as an adjunct to simulation training. If the review is intended to be comprehensive, there needs to be more details about the scope of AI in medical education, besides just as an adjunct to simulation training, such as the role of AI in providing diagnostic educational support, pattern recognition and as a teaching tool for training complex physiology. Subsequent to that, there can be an attempt to define feasibility.

Some suggestions:

  • Since the paper is about educating adult learners, there should be a basic section on adult learning theory and the nuances of it - Pedagogy vs Andragogy. 
  • Since identification of sepsis and crucial time sensitive steps for management are at the forefront of education, it would be worth looking into studies that have looked into the implementation of the pediatric sepsis education goals, if any.
  • You mention (line 60) "This review focuses on assessing current knowledge regarding pediatric sepsis education and the potential use of AI to improve clinical outcomes." I do not see where the later has been explored because the scope of AI is really broad when exploring patient clinical outcomes. Please elaborate or narrow to indicate the use of AI solely as a teaching modality.

I would suggest defining or explaining in more detail certain concepts that have been introduced:

  • Define sepsis
  • Richard Mayer's theory of multimedia learning and universal design for learning

-The paper references studies that implemented different strategies in LMIC countries but is there data for HIC? I think it is meaningful to review studies from HIC that have access to newer high fidelity resources and define their experience as well for a review paper. Following this there can be a comparison/defining limitations to the implementation of these tools in LMIC. 

Additionally, if the group is exploring barriers to pediatric sepsis education in LMIC, it might be worth highlighting briefly the limitations of a low/limited resource setting to management of septic shock and the need to tailor training for students training in these LMIC to adapt to resource limitation. 

Author Response

Reviewer 1

Comment 1: Thank you for the opportunity to review this paper.

Thank you for the feedback.

Comment 2: The paper aims to provide a comprehensive review of medical education specifically tailored to the teaching of pediatric sepsis and in the emergency room. However, the paper seems to read more for general medical training strategies than sepsis education specifically and I think the paper either needs to shift goals to define general education strategies and the role for AI or get more detailed with regards to sepsis education.

Thank you for the comment. We ‘ve tried our best to include specific pediatric sepsis sections in each section of the manuscript. We also added the following and believe it now flows better.

Lines 219-220: “This would include the development of clinical scenarios for pediatric sepsis, assessment of timely interventions and feedback on overall performance.”

Lines 234-236: “AI can assist in creating realistic clinical scenarios on pediatric sepsis and provide feedback regarding timely interventions. This would assist in assessing the time and adherence to sepsis guidelines in simulation scenarios.”

Lines 255-258: “This was also shown in a mixed methods study that used an AI-enabled VR simulator (VRS) and showed enhanced learning and improved multidisciplinary communication between nursing students and an AI medical doctor [37]. This would be key in pediatric sepsis education as communication is key in providing timely interventions.”

Comment 3: AI has essentially been introduced as an adjunct to simulation training. If the review is intended to be comprehensive, there needs to be more details about the scope of AI in medical education, besides just as an adjunct to simulation training, such as the role of AI in providing diagnostic educational support, pattern recognition and as a teaching tool for training complex physiology. Subsequent to that, there can be an attempt to define feasibility.

Thank you for the comment. We added the following in the manuscript.

Lines 219-220: “This would include the development of clinical scenarios for pediatric sepsis, assessment of timely interventions and feedback on overall performance.”

Lines 234-236: “AI can assist in creating realistic clinical scenarios on pediatric sepsis and provide feedback regarding timely interventions. This would assist in assessing the time and adherence to sepsis guidelines in simulation scenarios.”

Lines 255-258: “This was also shown in a mixed methods study that used an AI-enabled VR simulator (VRS) and showed enhanced learning and improved multidisciplinary communication between nursing students and an AI medical doctor [37]. This would be key in pediatric sepsis education as communication is key in providing timely interventions.”

Lines 240-242: In a recent study a structured approach using mind mapping and in situ simulation training in ED nurses resulted in statistically significant increase in identifying sepsis earlier and promoted the Hour-1 bundle treatment in sepsis patients [35].

Comment 4: Some suggestions:

Since the paper is about educating adult learners, there should be a basic section on adult learning theory and the nuances of it - Pedagogy vs Andragogy.

We have included the following sections to strengthen the educational value of AI in sepsis without expanding too much to avoid losing focus:

Lines 97-98: “These are all parts of pedagogic approach in which teachers decide what, how and when something will be learned [13].”

Lines 155-156: “Andragogy focuses on critical thinking and enhancement of knowledge in adult learners via application of knowledge in real life scenarios [13].”

Comment 5: Since identification of sepsis and crucial time sensitive steps for management are at the forefront of education, it would be worth looking into studies that have looked into the implementation of the pediatric sepsis education goals, if any.

We did not find specific pediatric sepsis studies on how education goals reflect in real life, which also makes our work timely and important. We added a limitation section and a comment on future studies and now reads as follows:

Lines 288-292: Limitations

“As a narrative review, no quantitative analysis was performed. Also, although pediatric sepsis educational goals have been suggested, there were no studies recorded evaluating how these goals changed pediatric sepsis educational landscape. Future studies should focus on providing data on the impact these educational goals had on pediatric sepsis education.”

Comment 6: You mention (line 60) "This review focuses on assessing current knowledge regarding pediatric sepsis education and the potential use of AI to improve clinical outcomes." I do not see where the latter has been explored because the scope of AI is really broad when exploring patient clinical outcomes. Please elaborate or narrow to indicate the use of AI solely as a teaching modality.

Thank you for the comment. We changed to the following in order to realign our manuscript to the use of AI as a teaching modality:

“This review focuses on assessing current knowledge regarding pediatric sepsis education and the potential use of AI to enhance pediatric sepsis education.”

Comment 7: I would suggest defining or explaining in more detail certain concepts that have been introduced:

Define sepsis

We inserted a figure 1 regarding sepsis definitions.

Richard Mayer's theory of multimedia learning and universal design for learning

We thank the reviewer – we have included the Richard Mayer’s theory with proper citations.

Lines 172-175: “For instance, Richard Mayer’s Theory of Multimedia Learning and Universal Design for Learning principles describe how to optimise teaching and learning by enhancing learner perception and comprehension through the way learning is presented [14,26,27].”

Comment 8: The paper references studies that implemented different strategies in LMIC countries but is there data for HIC? I think it is meaningful to review studies from HIC that have access to newer high fidelity resources and define their experience as well for a review paper. Following this there can be a comparison/defining limitations to the implementation of these tools in LMIC. Additionally, if the group is exploring barriers to pediatric sepsis education in LMIC, it might be worth highlighting briefly the limitations of a low/limited resource setting to management of septic shock and the need to tailor training for students training in these LMIC to adapt to resource limitation.

We thank the reviewer for the comment. We have  added the following section:

Lines 276-282: “On the other hand, high-income countries have access to high-fidelity simulation resources and infrastructure resources that offer realistic clinical scenarios enhancing clinical decision making and improve communication between multidisciplinary teams [41]. In a recent systematic review regarding the effectiveness of low-cost, technology-enhanced simulation training, data showed that low-cost, technology-enhanced simulation tools may serve as an efficient strategy. This may reduce disparities in access to high-quality training between high-income countries and LMICs [42].”

Reviewer 2 Report

Comments and Suggestions for Authors

The more effective use of technology in sepsis education is an exciting topic. My recommendations for the article are presented below:

1. Clinically distinguishing between heat and cold shock and selecting vasopressor drugs accordingly is no longer appropriate. The use of advanced monitoring methods is recommended. (Table 1: 12)
2. The advantages and disadvantages of educational methods are not specific to sepsis. It would be more appropriate to shorten these sections.
3. Data on the effectiveness of the clinical studies cited would be more interesting. Suggestions could be made on how these tools can be used more effectively in the future for prevention, diagnosis, treatment, prognostic assessment, and raising public awareness.

Author Response

Reviewer 2

Comment 1: The more effective use of technology in sepsis education is an exciting topic. My recommendations for the article are presented below:

We thank reviewer for positive feedback.

Comment 2:

Clinically distinguishing between heat and cold shock and selecting vasopressor drugs accordingly is no longer appropriate. The use of advanced monitoring methods is recommended. (Table 1: 12)

Thank you for the comment. Revised accordingly.

Comment 3:

The advantages and disadvantages of educational methods are not specific to sepsis. It would be more appropriate to shorten these sections.

Thank you for the comment. We shortened two sections deemed appropriate via team discussion. Lines 105-108 and 158-161.

Comment 4:

Data on the effectiveness of the clinical studies cited would be more interesting. Suggestions could be made on how these tools can be used more effectively in the future for prevention, diagnosis, treatment, prognostic assessment, and raising public awareness.

Thank you for the suggestion. The main goal of this review is to highlight AI tools as a teaching modality in sepsis to improve clinical outcomes. Future studies will elaborate on the effectiveness of the tools in clinical practice.

Reviewer 3 Report

Comments and Suggestions for Authors

This paper provides an insightful exploration of innovative training approaches for pediatric sepsis, a condition that continues to represent one of the most significant emergencies in pediatric medicine.

TITLE - OBJECTIVE: I think that the title and also the objective of the study should be modified. Indeed, the new perspectives are not only focused on AI as clearly stated in the manuscript. "Thematic analysis included the following categories:• Lectures,• Seminars,• E-learning,• Hybrid learning,• Technology-enhanced learning (gamification, virtual reality (VR) and augmented reality 76 (AR) simulation,• Artificial intelligence,• Barriers to implementation of artificial intelligence in sepsis education"

INTRODUCTION: it would better specified if there is a gap of knoweldge that need to be addressed.

Page 2, Line 4. the decrease of mortality should be ideally defined.

FIGURE: The figure could include plus and minus of the single methods of training. 

TABLE: The authors could consider adding a summary table outlining the main studies investigating newer training methods. For clarity, the table could include three columns detailing: (1) the training methods used, (2) the study outcomes assessed, and (3) the main results.

Author Response

Reviewer 3

Comment 1: This paper provides an insightful exploration of innovative training approaches for pediatric sepsis, a condition that continues to represent one of the most significant emergencies in pediatric medicine.

We thank the reviewer for the feedback.

Comment 2: TITLE - OBJECTIVE: I think that the title and also the objective of the study should be modified. Indeed, the new perspectives are not only focused on AI as clearly stated in the manuscript. "Thematic analysis included the following categories:• Lectures,• Seminars,• E-learning,• Hybrid learning,• Technology-enhanced learning (gamification, virtual reality (VR) and augmented reality 76 (AR) simulation,• Artificial intelligence,• Barriers to implementation of artificial intelligence in sepsis education"

Updated aims and gap of knowledge focus of the review

Lines 60-61: “This review focuses on assessing current knowledge regarding pediatric sepsis education and discussing the potential use of AI to improve clinical outcomes enhance pediatric sepsis education.”

Comment 3: INTRODUCTION: it would better specified if there is a gap of knowledge that need to be addressed.

Updated aims and gap of knowledge focus of the review. See above comment.

Comment 4: Page 2, Line 4. the decrease of mortality should be ideally defined.

Updated and now reads as follows:

“with a decrease in 30 day all-cause in-hospital mortality”.

Comment 5: FIGURE: The figure could include plus and minus of the single methods of training. 

Updated in all figures.

Comment 6: TABLE: The authors could consider adding a summary table outlining the main studies investigating newer training methods. For clarity, the table could include three columns detailing: (1) the training methods used, (2) the study outcomes assessed, and (3) the main results.

Thank you for this comment. Although we contemplated the creation of such table, it would be very large and difficult for the reader to follow. We didn’t identify another way to present these data, but we believe that the text is comprehensive yet succinct and we hope that it can clearly communicate the findings of the cited studies.